# Analysis of the Specific Immune Response after the Third Dose of mRNA COVID-19 Vaccines in Organ Transplant Recipients: Possible Spike-S1 Reactive IgA Signature in Protection from SARS-CoV-2 Infection

**DOI:** 10.3390/microorganisms10081563

**Published:** 2022-08-03

**Authors:** Monica Miele, Rosalia Busà, Giovanna Russelli, Maria Concetta Sorrentino, Mariangela Di Bella, Francesca Timoneri, Giampiero Vitale, Elisa Calzolari, Patrizio Vitulo, Alessandra Mularoni, Pier Giulio Conaldi, Matteo Bulati

**Affiliations:** 1Research Department, Mediterranean Institute for Transplantation and Advanced Specialized Therapies (IRCCS ISMETT), 90127 Palermo, Italy; rbusa@ismett.edu (R.B.); grusselli@ismett.edu (G.R.); mdibella@fondazionerimed.com (M.D.B.); ftimoneri@fondazionerimed.com (F.T.); givitale@fondazionerimed.com (G.V.); pgconaldi@ismett.edu (P.G.C.); mbulati@ismett.edu (M.B.); 2Ri.MED Foundation, 90133 Palermo, Italy; 3Department of Laboratory Medicine and Advanced Biotechnologies, Mediterranean Institute for Transplantation and Advanced Specialized Therapies (IRCCS ISMETT), 90127 Palermo, Italy; msorrentino@ismett.edu; 4Allergy and Respiratory Diseases, Department of Internal Medicine (DIMI), IRCCS Policlinico San Martino, University of Genoa, 16124 Genoa, Italy; ecalzolari@ismett.edu; 5Department for the Treatment and Study of Cardiothoracic Diseases and Cardiothoracic Transplantation, Mediterranean Institute for Transplantation and Advanced Specialized Therapies (IRCCS ISMETT), 90127 Palermo, Italy; pvitulo@ismett.edu; 6Department of Infectious Diseases, Mediterranean Institute for Transplantation and Advanced Specialized Therapies (IRCCS ISMETT), 90127 Palermo, Italy; amularoni@ismett.edu

**Keywords:** mRNA vaccine, solid organ transplant recipients, immune response, IgG, IgA, T cell response, SARS-CoV-2, COVID-19

## Abstract

**Background:** Several studies have indicated that anti-SARS-CoV-2 mRNA vaccinations are less effective in inducing robust immune responses among solid organ transplant recipients (SOTRs) compared with the immunocompetent. The third dose of vaccine in SOTRs showed promising results of immunogenicity, even though clinical studies have suggested that immunocompromised subjects are less likely to build a protective immune response against SARS-CoV-2 resulting in lower vaccine efficacy for the prevention of severe COVID-19. **Methods:** Serological IgG and IgA were analyzed through CLIA or ELISA, respectively, while Spike-specific T cells were detected by ELISpot assay after the second and third dose of vaccine in 43 SOTRs. **Results:** The third dose induced an improvement in antibody response against SARS-CoV-2. We also reported a strong correlation between specific humoral and cellular responses after the third dose, even though we did not see significant changes in the magnitude of the SARS-CoV-2-specific T cell response. SOTRs who contracted the SARS-CoV-2 infection after the third dose, despite eliciting a positive IgG response, failed to mount an anti-Spike-S1 IgA response, both after the third dose and after SARS-CoV-2 infection. **Conclusions:** We can conclude that serum IgA detection can be helpful, along with IgG detection, for the evaluation of vaccine efficacy, principally in fragile subjects at high risk of infection.

## 1. Introduction

Coronavirus disease (COVID-19) is caused by a complex interaction between a severe acute respiratory syndrome coronavirus 2 (SARS-CoV-2) infection and the host immune response. Observational studies on vaccine effectiveness have demonstrated the efficacy of mRNA vaccines, including Pfizer-BioNTech BNT162b2 and Moderna mRNA-1273, in the prevention of COVID-19-associated hospitalization and severe disease among immunocompetent people [1,2,3,4]. However, several studies have indicated that anti-SARS-CoV-2 mRNA vaccination is less effective in inducing robust immune responses among solid organ transplant recipients (SOTRs) on immunosuppressive medication regimens [5,6,7,8,9]. Indeed, studies have reported [9,10,11] that more than 60% of SOTRs fail to elicit humoral and cellular immunity after two doses of mRNA vaccines. The suboptimal immunogenicity of vaccines in SOTRs after transplantation has already been demonstrated for influenza due to co-morbidity conditions and the effects of the immunosuppressive drugs used [12,13,14]. A systematic review [15] reported that, among SOTRs, after the mRNA primary vaccination cycle (two doses), non-responder rates ranged from 18 to 100% (35–98% in kidney transplant recipients, 19–63% in liver transplant recipients, 25–88% in heart transplant recipients, and 59–100% in lung transplant recipients). Reports on the third dose in SOTRs showed promising results of immunogenicity [16,17] even though some clinical studies have suggested that immunocompromised subjects are less likely to build a protective immune response against SARS-CoV-2 [5,9,18,19,20] resulting in lower vaccine efficacy for the prevention of severe COVID-19 [21,22]. In this study, we analyzed humoral and cellular immune responses after the second and third dose of mRNA vaccine and the different factors associated with the mounting of SARS-CoV-2-specific immunity in a cohort of 43 SOTRs. Moreover, we recorded the clinical outcomes of patients who contracted the infection after the third vaccine dose. Our results highlight that the additional third dose resulted in a markedly improved antibody response against SARS-CoV-2. We also reported a strong correlation between specific humoral and cellular responses after the third dose, even though we did not see significant changes in the magnitude of the SARS-CoV-2-specific T cell response. Nevertheless, more than one-third of our cohort of SOTRs contracted the SARS-CoV-2 infection after the third dose of mRNA vaccine. Interestingly, we reported that this group of infected–vaccinated SOTRs, despite eliciting a good SARS-CoV-2-specific IgG response after the third dose, failed to mount an anti-Spike-S1 IgA response, either before or after SARS-CoV-2 infection. These results lead us to hypothesize a possible protective role of serum IgA in the prevention of COVID-19. For this reason, we suggest that serum IgA detection can be helpful, alongside IgG detection, for the evaluation of vaccine efficacy, principally in fragile subjects at high risk of infection, such as SOTRs.

## 2. Materials and Methods

### 2.1. Recruitment and Clinical Sample Collection 

We prospectively enrolled the first 43 SOTRs received by ISMETT Hospital for post-transplant follow-up, who had received the second dose of mRNA (Pfizer-BioNTech or Moderna) vaccine between January and May 2021 and the third dose between September and December 2021, and from whom peripheral blood mononuclear cells (PBMCs) and serum samples were serially collected. Among these 43 patients, 14 presented COVID-19-like symptoms and tested positive for SARS-CoV-2 RT-PCR (nasopharyngeal/oral swab) after the third dose of the vaccine. Five patients required hospitalization, while nine patients had mild or moderate symptoms and were treated for the infection at home. Characteristics of the population are summarized in Table 1. None of the patients included in the study had a history of PCR-confirmed SARS-CoV-2 infection, neither before nor after the second dose of the vaccine. To monitor infection during the overall period of follow-up, apart from a positive nasopharyngeal swab, we also evaluated the presence of an immune response against the N protein. The anti-N response was determined with the ARCHITECT Quant test (Abbott) using the chemiluminescent assay anti-SARS-CoV-2-N-domain CMIA (IgG and IgM) (Abbott) and SARS-CoV-2 ELISpot against N-peptides mix (see ELISpot paragraph) 3 weeks after the second (T1) and the third dose (T2) (data not shown). Blood, PBMCs, and serum samples were collected at T1 and T2 for the analysis of humoral and cellular responses. We also collected serum samples from SARS-CoV-2-infected SOTRs 1 month after nasopharyngeal/oral swab negativization (T3). The study was approved by the IRCCS ISMETT Institutional Research Review Board (IRRB/00/21) and by the Ethics Committee of ISMETT and all enrolled patients signed the written informed consent form.

### 2.2. Detection of SARS-CoV-2 Anti-Spike Immunoglobulins

Anti-Spike IgG and IgA were detected in sera from enrolled subjects. The chemiluminescent immunoassay (CLIA) LIAISON^®^ SARS-CoV-2 S1/S2 IgG (DiaSorin S.p.A., Saluggia, VC, Italy) was used to perform quantitative detection of IgG antibodies against S1 and S2 fragments of the Spike protein. The test was used on the fully automated LIAISON^®^ XL Analyzer (DiaSorin S.p.A., Saluggia, VC, Italy). The SARS-CoV-2 S1/S2 IgG antibody concentrations were expressed as Binding Antibody Unit (BAU) per ml (BAU/mL), and values > 33.8 BAU/mL were considered positive. The anti-SARS-CoV-2 IgA (EUROIMMUN, PerkinElmer Company, Hong Kong, China) enzyme-linked immunoassay (ELISA) was used for the semi-quantitative detection of IgA antibodies to S1 fragments of the viral surface Spike protein. The test was used on the fully automated EUROIMMUN Analyzer I (EUROIMMUN, PerkinElmer Company). The anti-SARS-CoV-2 ELISA IgA antibody concentrations were expressed as the ratio of the extinction of the sample to that of the calibrator, and the results were graded as follows: Negative (<0.8), Equivocal (≥0.8 to 1.1), and Positive (>1.1). 

### 2.3. SARS-CoV-2 ELISpot Assay

ELISpot assay was used to detect T cell response. We isolated PBMCs from whole blood of studied subjects by density gradient centrifugation using a cell preparation tube with sodium citrate (BD Vacutainer^®^ CPT™) according to the manufacturer’s protocol. PBMCs were counted using the Sysmex XN-2000™ Hematology System. Human IFN-γ ELISpot plus kit (Mabtech AB, Stockholm, Sweden) were used to detect IFN-γ-secreting T cells, according to the manufacturer’s protocol. The assay was performed in duplicate, stimulating 2.5 × 10^5^ ± 0.5 × 10^5^ PBMCs/mL for 20–22 h, at 37 °C in a 5% CO_2_ humidified atmosphere, with 1 µg/mL overlapping peptides spanning the SARS-CoV-2 Spike (Mix I and II, respectively, of 158 and 157 peptides, purity > 90% derived from a peptide scan, 15 mers with 11 aa overlap) or an N protein peptide pool (purity > 90%, JPT Peptide Technologies, Berlin, Germany). The PBMCs were cultured in RPMI 1640 medium (BIOWEST, Nuaillé, France), supplemented with 5% GemCell™ U.S. Origin Human Serum AB (BIOIVT, Westbury, NY, USA) and 1% L-Glutamine (Euroclone, Pero, Italy). A negative control (RPMI + 5% Human Serum AB) and positive controls, such as anti-CD3 and CEFX PepMix (a pool of 176 known peptides from various infectious agents, JPT Peptide Technologies, Germany) were also included. The number of SARS-CoV-2-specific IFN-γ-secreting cells was measured, according to ELISpot guidelines [23], with an ELISpot Reader (Autoimmun Diagnostika (AID) GmbH, Straßberg, Germany) and using ELISpot Software (AID). Mean spot counts for negative control wells were subtracted from the mean of test wells, and the spots are presented as Spot Forming Unit (SFU) per million PBMCs to generate normalized readings. To determine the lower limit for indicating a positive response (cutoff), we considered the mean value of responses of unstimulated wells plus two standard deviations (SDs) (cutoff = 112 SFC/10^6^ PBMC).

### 2.4. Statistical Analysis

Statistical analysis was performed with GraphPad Prism 8. The Wilcoxon matched-pairs signed-rank test was used to compare paired nonparametric data. Correlations were performed using the Spearman’s rank correlation coefficient. Statistical significance was determined as * *p* < 0.05, ** *p* < 0.01, *** *p* < 0.001, and **** *p* < 0.0001.

## 3. Results

### 3.1. Characteristics of the Study Population

In this study, 43 SOTRs (26 men, 60%) with a median age of 54 years (IQR, 46–62.3 years), with no history of COVID-19 and a negative SARS-CoV-2 anti-N serology at the time of inclusion, were enrolled. Patients’ baseline characteristics are shown in Table 1. The IS therapy included calcineurin inhibitors (CNI, tacrolimus) (93%, 40/43 patients), mTOR inhibitors (everolimus) (7%, 3/43 patients), Mycophenolate-mofetil (MMF) (69.8%, 30/43 patients), and steroids (53.5%, 23/43 patients).

### 3.2. The Third Dose of Pfizer-BioNTech BNT162b2 mRNA Vaccine Induces an Improvement in Humoral Response

Serum Spike-specific IgG antibody titers were compared after the second (T1) and third dose (T2) of mRNA vaccine in SOTRs. The proportion of SOTRs with a positive serologic response was 56% at T1 and 84% at T2. As reported in Figure 1a, we found a significant increase in the median value of Spike-specific IgG after the third dose (2.6-fold change T1/T2), suggesting that an additional dose provides an efficient boost of antibody response against SARS-CoV-2. Indeed, at T1 the median value of specific IgG was 335.40 BAU/mL (IQR, 127.4–811.2; SEM = 101.4), while a median value of 872.0 BAU/mL was reached at T2 (IQR, 331.2–2109.0; SEM = 263.6; *p* < 0.0001). Thirteen of the 19 patients (68%) who were seronegative (anti-S IgG < 33.8 BAU/mL) at T1 showed seroconversion after the third dose. Moreover, to better characterize the serological response we evaluated the serum Spike-specific IgA antibody levels at the same time point. As depicted in Figure 1b, we observed a slight but significant increase in the median value of specific IgA after the third dose (1.4-fold change T1/T2). Indeed, at T1 the median ratio of specific IgA was 1.0 (IQR, 0.40–4.66; SEM = 0.52), and at T2, 1.4 (IQR, 0.45–7; SEM = 0.59; *p* = 0.031). Four of the 15 patients (27%) who were seronegative (anti-S IgA < 1.1) at T1 showed seroconversion after the third dose.

### 3.3. The Magnitude of Vaccine-Specific T Cell Responses

Conversely, we observed that the third dose did not induce an increase in the cellular response in SOTRs. Indeed, we noticed a similar trend in Spike-elicited IFN-γ release in the SOTRs, where the spot median value was 129.5 (IQR, 69–254; SEM = 37.3) and 129 (IQR, 49–312; SEM = 39) SFC per 10^6^ PBMCs cells at T1 and T2, respectively (*p* = 0.694) (Figure 1c). Particularly, at T1 65% (28/43) of patients mounted a positive T cell response, while at T2 we reported a slight reduction (53%, 23/43) in patients with a positive T cell response. Interestingly, 23/28 SOTRs (82%), who positively raised viral immunity at T1, maintained a Spike-specific T cell response at T2, and two patients without a T cell response at T1, mounted a positive response at T2. As a control, we used CEFX immuno-dominant peptide pools to stimulate PBMC, obtaining similar responses at T1 and T2 (data not shown).

### 3.4. T Cell Response Correlates with IgG Antibody Levels after the Third Vaccination Dose

To better understand whether humoral and cellular responses induced by mRNA vaccines have interdependent effects, we assessed the correlation magnitude existing between the cellular responses against IgG antibody levels after vaccination. At T1 we found a significant correlation between T cell response and IgG (r = 0.5964, *p* < 0.0001) (Figure 2a). At T2 we observed a strong significant correlation comparing T cell response with IgG serum levels (r = 0.9998, *p* < 0.0001) (Figure 2b). Moreover, as the suboptimal immunogenicity of vaccines in SOTRs due to immunosuppressive (IS) drug treatments is known, we evaluated the correlation between immune responses and IS regimens. Comparing humoral or T cell response versus type, dosage, and level of immunosuppressive (IS) drugs, we did not find any significant correlations (data not shown).

### 3.5. Clinical Outcome after the Third Dose

In order to define the real-life efficacy of the third dose of an mRNA vaccine on SOTRs, we evaluated the correlation between immune response and clinical outcomes, such as the severity of disease, hospitalization, and death after SARS-CoV-2 infection. During follow-up, at a median time of 130 days (IQR, 120–169.5 days) after the third dose, 14/43 SOTRs (32% of total) enrolled in our study had contracted SARS-CoV-2 infection. The clinical characteristics of patients are described in Table 2. The virus-infected SOTRs had a median age of 53 (IQR, 41–58 years) and were six lung, three liver, three kidney, and two heart recipients, with a median time from transplant of 7 years. Nine of them showed mild to moderate symptoms, including fever, pain, cough, and asthenia, which did not need hospitalization, but 1/9 of patients were treated with an antiviral drug to prevent complications. Five SOTRs showed severe symptoms that required hospitalization: 3/5 recovered and 2/5 had a fatal outcome related to COVID-19. The two deceased patients, TxVAC11 and TxVAC19, had been admitted to ICU, the former was treated for ARDS with extracorporeal membrane oxygenation (ECMO), and the latter, who had been previously admitted for ischemic stroke, was treated with non-invasive ventilation. Among the nine patients with mild symptoms who did not require hospitalization, five had an anti-SARS-CoV-2 positive immune response of both T cells and serum IgG, three had either an IgG or T cell positive response, and one showed no detectable immunological response and was treated with antiviral drugs. Among the three hospitalized patients who recovered, two showed only the positive IgG response and were treated with monoclonal antibodies, while one showed only the T cell response and needed antiviral treatment. Finally, the two deceased patients did not show any humoral or T cell responses, neither after the second nor the third dose. Concerning Spike-specific IgA, we reported a positive serology only in two non-hospitalized patients who recovered, which also presented both IgG and T cell positive responses. Amongst the remaining 29 SOTRs who did not contract the infection, almost all of them mounted at least one type of immune response. Given the heterogeneity of the anti-SARS-CoV-2 immune response after the third vaccination, apart from the two deceased infected patients, we decided to compare the humoral assessment in uninfected– and infected–vaccinated SOTRs to identify predictive biomarkers of clinical outcomes. Interestingly, comparing serum IgG levels at T2, we showed that both groups had a positive specific IgG response with no significant statistical difference (*p* = 0.5270). As depicted in Figure 3a, the median value in uninfected SOTRs was 872.0 BAU/mL (IQR, 486.7–1927.0; SEM = 347.0), while SARS-CoV-2-infected SOTRs had a median value of 987.0 BAU/mL (IQR, 45.63–2123.0; SEM = 415.9). Surprisingly, after the third dose, we observed that the uninfected SOTRs had a significant IgA positive serology (*p* = 0.0054) with a median ratio of 5.4 (IQR, 0.46–7.0; SEM = 0.7), while SOTRs who contracted the infection had a negative IgA serology with a median ratio of 0.56 (IQR, 0.09–1; SEM = 0.4) (Figure 3b).

### 3.6. Humoral Assessment after Breakthrough Infection

Finally, comparing the humoral responses in infected–vaccinated SOTRs at T2 and after SARS-CoV-2 infection (T3), we found a non-significant decrease (*p* = 0.9705) in the median value of specific IgG after SARS-CoV-2 infection (1.50-fold change), which was 888.9 BAU/mL (IQR, 0.00–2239; SEM = 564.2) at T2 and 590.7 BAU/mL (IQR, 115.5–2332; SEM = 531.5) at T3 (Figure 4a). Concerning serum IgA, the median ratio value was 0.51 (IQR, 0.045–1000; SEM = 0.4160) at T2 and 0.8850 (IQR, 0.5750–1.375; SEM = 0.4) at T3 (Figure 4b). These results suggest that, in the SOTR subgroup who contracted the infection after the third dose, the natural infection did not sufficiently boost either an IgG or IgA humoral response against SARS-CoV-2. Importantly, while these subjects maintain a positive IgG response after three doses of vaccine, they seem to have an impaired ability to mount an anti-Spike-specific IgA response even after the viral infection.

## 4. Discussion

A deep understanding of the clinical protection provided by mRNA vaccines in immunocompromised populations, such as SOTRs, is crucial for several reasons, including: making decisions on the prophylactic use of anti-SARS-CoV-2 monoclonal antibodies during hospitalization or the use of antiviral therapies; advising on the use of preventative strategies, such as mask wearing even after vaccination; and providing recommendations for additional vaccine booster doses [24]. The initial recommendations for people with immunocompromised conditions were two doses of an mRNA COVID-19 vaccine. This recommendation was updated to three doses in the second half of 2021, with a fourth dose currently recommended at least 3 months after the third dose as a vaccine booster. In this study, the third dose of the mRNA vaccine was evaluated in 43 SOTRs, who had received the second dose approximately 4 months previously. After 3–5 weeks from the third dose of the vaccine, we reported a significant improvement in anti-SARS-CoV-2 IgG and IgA. Regarding Spike-specific cellular immune response, we did not observe an improvement after the third dose. In a recent study, however, it was demonstrated that SOTRs receiving the third dose after 2 months from the second dose showed an improved Spike-specific T cell response [25]. The reason we did not observe a significant improvement in the T cell response in our SOTRs can be due to the time elapsed between the second and third doses of the vaccine, which was greater than 3 months, since it has been described that SARS-CoV-2-specific T cells decline with a half-life of 3 to 5 months after natural infection or first vaccination cycle [26]. Nevertheless, the third booster dose induced an increase from 77 to 90% in total response in SOTRs, merging humoral and cellular immune responses. This result is also confirmed by the increased significant correlation magnitude existing between the cellular response and the IgG antibody levels. Moreover, in our cohort of SOTRs, the type of immunosuppressive regimen does not appear to have a significant impact on immune response. During follow-up, none of the SOTRs included in the study reported infection until after the administration of the third dose. At a median time of 130 days from the third dose, during the Omicron wave, 14/43 (32%) patients contracted the infection. Recently, it was reported that even with three doses of mRNA vaccine, SOTRs show a suboptimal ability to respond to the Omicron variant compared with previous variants (from wild type to delta), which, in part, may explain the high numbers of vaccine breakthrough infections [27,28]. Among the 14 infected–vaccinated SOTRs, nine did not need hospitalization and showed mild to moderate symptoms, three SOTRs showed severe symptoms that required hospitalization and subsequently recovered, and two had a fatal outcome. The remaining 29 vaccinated SOTRs who did not contract the infection showed positivity for both IgG and IgA anti-SARS-CoV-2 antibodies. Despite maintaining anti-SARS-CoV-2 IgG positivity similar to their uninfected–vaccinated counterparts, almost all infected–vaccinated SOTRs were negative for anti-Spike IgA. Surprisingly, these subjects did not show a positive IgA serology even after natural infection. IgA antibodies, which account for 10 to 15% of human immunoglobulins [29], are known to play a central role in mucosal immunity, which is important in protection against respiratory infections, such as SARS-CoV-2. It was recently demonstrated that the presence of serological IgA is superior to IgM as an early serological marker of recent SARS-CoV-2 infection, demonstrating the importance of IgA in the disease progression [30,31] and in the prediction of the clinical course of COVID-19 [32]. In addition, we recently reported that both the first vaccination cycle and the booster dose of the anti-SARS-CoV-2 mRNA vaccine can induce a strong anti-Spike-specific serological IgA response in immunocompetent subjects [33]. Our results on infected–vaccinated SOTRs, together with these recent findings, lead us to speculate that these subjects are probably more susceptible to contracting the infection compared with those who are not infected, who show a positive IgA serology. For this reason, we suggest that serum IgA detection can be helpful, along with the detection of IgG, for the evaluation of vaccine efficacy, mainly in fragile subjects at high risk of infection, such as SOTRs. The limitations of our study are the small cohort of SOTRs studied, and the heterogeneity in the type of transplanted organs and of the immunosuppressive therapies.

## 5. Conclusions

We can conclude that, despite three doses of vaccine, SOTRs remain at risk of severe COVID-19 supporting the necessity for continued efforts to limit the risk of COVID-19 in this population [24]. The fourth dose of mRNA COVID-19 vaccines is now recommended for immunocompromised individuals. Future studies will be important for understanding the effectiveness of the fourth dose in preventing hospitalization, the residual risk of severe COVID-19 among SOTRs, and the durability of protection. Additional booster doses or modified vaccines, through the addition of adjuvants or increasing antigens [24], can provide the benefit of overcoming the impaired immune response observed in SOTRs. Finally, taking additional precautions to mitigate the risk of SARS-CoV-2 exposure among SOTRs, such as constant monitoring of the SARS-CoV-2-specific immune response, remain of crucial importance.

## Figures and Tables

**Figure 1 microorganisms-10-01563-f001:**
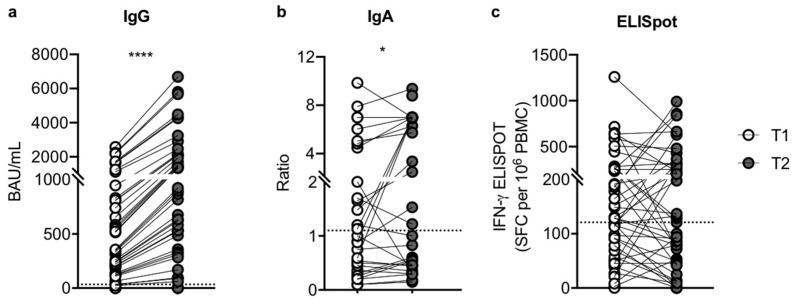
Humoral and cellular immune response to SARS-CoV-2 vaccination in SOTRs (n = 43) after the second (T1) and third (T2) dose of Pfizer-BioNTech BNT162b2 vaccine. (**a**) Comparison of anti-SARS-CoV-2 S1/S2 IgG concentration between T1 (white dots) and T2 (grey dots). Samples with anti-SARS-CoV-2 S1/S2 IgG concentration >33.8 BAU/mL were considered positive. (**b**) Comparison of anti-SARS-CoV-2 S1 IgA ratio between T1 (white dots) and T2 (grey dots). Ratios ≥ 1.1 were considered positive. (**c**) T cell responses (IFN-γ ELISpot SFC per 10^6^ PBMC) to Spike were compared between T1 (white dots) and T2 (grey dots) in SOTRs. Each dot plot represents the normalized mean spot count from duplicate wells (2.5 ± 0.5 × 10^5^ PBMC/well) for each subject after subtraction of the spot count of unstimulated cells. IFN-γ ELISpot > 112 SFC/10^6^ PBMC were considered positive. The connection lines represent the antibody value of each subject at T1 and T2, while the dotted line corresponds to the threshold. The significance was determined using the Wilcoxon matched-pairs signed-rank test (two-tailed), **** *p* < 0.0001, * *p* = 0.0314.

**Figure 2 microorganisms-10-01563-f002:**
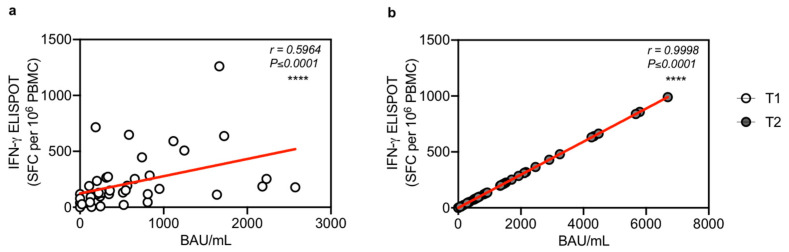
Correlation of IgG humoral responses against cellular immune response to SARS-CoV-2 vaccination in SOTRs (n = 43) after second (T1) and third (T2) dose of Pfizer-BioNTech BNT162b2 vaccine. (**a**) Correlation of T cell responses (IFN-γ ELISpot SFC per 10^6^ PBMC) to Spike against anti-SARS-CoV-2 IgG (BAU/mL) at T1 (white dots). (**b**) Correlation of T cell responses (IFN-γ ELISpot SFC per 10^6^ PBMC) to Spike against anti-SARS-CoV-2 IgG (BAU/mL) at T2 (grey dots). Spearman’s rank correlation (two-sided) was used to test the significance and the *p*-values and r-values (correlation coefficient) are indicated for each panel, **** *p* < 0.0001.

**Figure 3 microorganisms-10-01563-f003:**
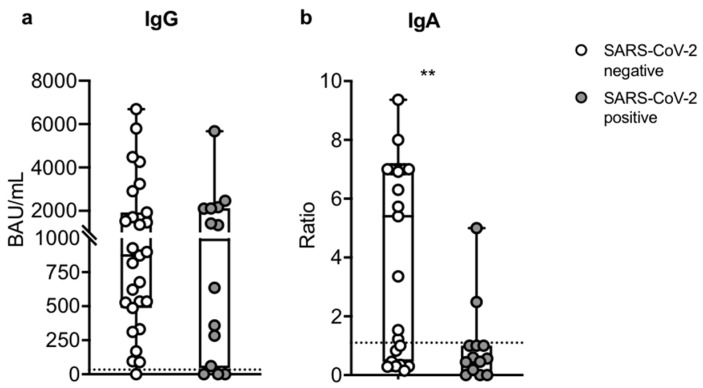
Humoral assessment in uninfected– (n = 27) and infected–vaccinated SOTRs (n = 14) after the third dose of mRNA vaccine. (**a**) Comparison of IgG Spike (BAU/mL) in uninfected– (white dots) and infected–vaccinated SOTRs (grey dots) after the third booster dose. (**b**) Comparison of IgA Spike Ratio in uninfected– (white dots) and infected–vaccinated SOTRs (grey dots) after the third booster dose. The dotted line corresponds to the threshold. The significance was determined using the Mann–Whitney U test, ** *p* = 0.0054.

**Figure 4 microorganisms-10-01563-f004:**
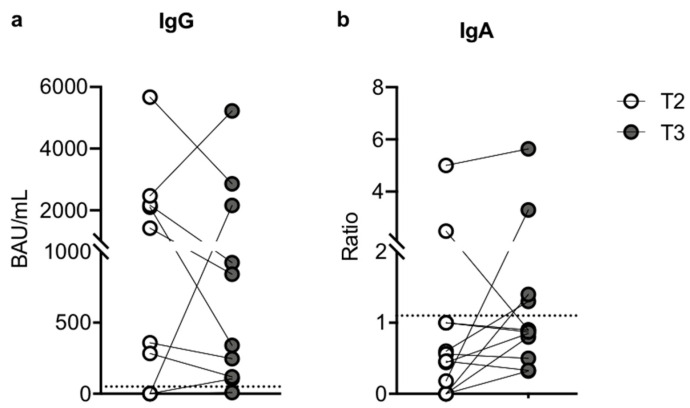
Comparison of humoral responses in infected–vaccinated SOTRs at T2 and after SARS-CoV-2 infection (T3) (n = 12). (**a**) Comparison of anti-SARS-CoV-2 S1/S2 IgG concentration between T2 (white dots) and T3 (grey dots). Samples with anti-SARS-CoV-2 S1/S2 IgG concentration > 33.8 BAU/mL were considered positive. (**b**) Comparison of anti-SARS-CoV-2 S1 IgA Ratio between T2 (white dots) and T3 (grey dots). Ratios ≥ 1.1 were considered positive. The connection lines represent the antibody value of each subject at T1 and T2, while the dotted line corresponds to the threshold. The significance was determined using the Mann–Whitney *U* test.

**Table 1 microorganisms-10-01563-t001:** Baseline characteristics of SOTRs.

Variable	SOTRs (n = 43)
Age, mean yr (SD)	54.2 (12.7)
Gender, M (%)	26 (60.4)
Type of transplant, n (%)	
*kidney*	11 (25.6)
*lung*	14 (32.6)
*liver*	10 (23.2)
*heart*	7 (16.3)
*liver-kidney*	1 (2.3)
Time from transplant, median yr (range)	6 (1–27)
Immunosuppressive treatment, n (%),	
*Calcineurin inhibitors ^1^ (3.27 to 12.43 ng/mL) mean ng/mL (SD)*	40 (93), 7.2 (2.0)
*mTOR inhibitors ^2^ (2.43 to 5.47 ng/mL) mean ng/mL (SD)*	3 (7), 3.6 (1.60)
*Mycophenolate-mofetil (MMF) (180 to 2000 mg)* *mean mg (SD)*	30 (69.7), 968.5 (487.6)
*Steroids (1.25 to 12.18 mg) mean mg (SD)*	23 (53.5), 5.4 (2.4)
Timespan between T1/T2, mean days (range)	187.7 (114–244)
Timespan between T1/sampling, mean days (range)	36.51 (15–136) ^3^
Timespan between T2/sampling, mean days (range)	40.3 (11–132) ^3^
Comorbidities, n (%)	
*Diabetes*	9 (20.93)
*Obesity*	7 (16.27)
*Hypertension*	17 (39.53)
*Dyslipidaemia*	8 (18.60)
*Active or previous smoke*	14 (32.56)
*Cardiovascular disease*	12 (27.90)
*Kidney disease*	6 (13.95)
*Pulmonary disease*	6 (13.95)
*Gastrointestinal disease*	12 (30.23)
*Endocrinal disease*	7 (16.27)
*Hepatopancreatic disease*	3 (6.97)
*History of malignancy*	16 (37.20)

^1^ tacrolimus; ^2^ everolimus; ^3^
*p*-value between timespan between T1/sampling and timespan between T2/sampling was not significant (=0.480). Abbreviations: yr, year; SD, standard deviation; M, male; MMF, Mycophenolate-mofetil.

**Table 2 microorganisms-10-01563-t002:** Baseline characteristics, humoral and cellular immune response and COVID-19 outcome in 14 SOTRs. SOTRs with a positive humoral or cellular immune response are highlighted in bold. Abbreviations: TX: transplant; MMF: Mycophenolate-mofetil; aVT: antiviral therapy; mAb: monoclonal antibody.

Basal Characteristics				Immunosuppressive Therapy			Immune Response after 3rd Dose (T2)			Clinical Presentation		
Patient (n = 14)	Age	Tx	Years since Tx	Calcineurin INHIBITORS (ng/mL)	MMF (mg/die)	Steroid (mg/die)	Anti-Spike IgG (BAU/mL)	Anti-Spike IgA (Ratio)	T Cell Response (SFC/million PBMC)	Symptoms	Treatment	Outcome
TxVAC1	65	liver	27	--	1000		21.5	0.6	**311**	severe	aVT, steroid	Hospitalized, discharged
TxVAC8	53	lung	8	12.4	2000	5	**220.9**	0.9	**320**	moderate		Nonhospitalized, recovered
TxVAC9	69	lung	1	10.2	1000	10	**73.3**	0.44	0	severe	mAb	Hospitalized, discharged
TxVAC11	41	kidney	3	6.8	720	5	6.8	0	0	severe/ critical	mAb, ECMO	Hospitalized, died
TxVAC19	70	kidney	1	6.9	1080	--	18.1	0	0	severe/ critical	aVT, mAb	Hospitalized, died
TxVAC25	41	lung	5	10.9	2000	5	20.9	0.9	0	moderate	aVT	Nonhospitalized, recovered
TxVAC26	20	lung	6	10.2	1000	5	21.6	1	**198**	moderate	-	Nonhospitalized, recovered
TxVAC28	58	kidney	18	6.4	1000	-	**170.8**	1.03	**312**	moderate	-	Nonhospitalized, recovered
TxVAC48	50	lung	3	7.2	500	5	**82.9**	0.56	9	severe	mAb	Hospitalized, discharged
TxVAC66	49	liver	3	5	250	5	**166.4**	0.18	**839**	moderate	-	Nonhospitalized, recovered
TxVAC68	53	liver	11	4.2	1000	-	**111**	0.46	42	mid	-	Nonhospitalized, recovered
TxVAC74	59	heart	12	6.3	1000	-	**938.6**	**2.9**	**210**	mild	-	Nonhospitalized, recovered
TxVAC83	40	heart	12	6.4	1000	-	**>1040**	**5.6**	**365**	moderate	-	Nonhospitalized, recovered
TxVAC85	51	lung	25	8.8	1000	5	**>1040**	0.8	94	moderate	-	Nonhospitalized, recovered

## Data Availability

The original contributions presented in the study are included in the article. Further inquiries can be directed to the corresponding authors.

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
