# Peer review of "Analysis of the Specific Immune Response after the Third Dose of mRNA COVID-19 Vaccines in Organ Transplant Recipients: Possible Spike-S1 Reactive IgA Signature in Protection from SARS-CoV-2 Infection"

_microorganisms, 2022, doi:10.3390/microorganisms10081563_

Round 1

Reviewer 1 Report

The manuscript by Monica et al., reports the importance of serum IgA levels for protection against breakthrough SARS-CoV-2 infections in solid organ transplant recipients. 

(line 160) 3.2: Is this only related to the Pfizer vaccine? what about the moderna vaccine?

The study is conducted very well and data is analyzed and presented clearly. But, the major concern is difficulty in reading the manuscript due to poor English proficiency.  Highly recommend the manuscript be edited extensively by an English professional for further consideration.

Author Response

The manuscript by Monica et al., reports the importance of serum IgA levels for protection against breakthrough SARS-CoV-2 infections in solid organ transplant recipients. 

We thanks very much the reviewer for the positive comments and the useful suggestions.

I wish to report that the manuscript is by Miele et al. (Miele is the surname)

 (line 160) 3.2: Is this only related to the Pfizer vaccine? what about the moderna vaccine?

  1. The results obtained are related to both type of mRNA vaccine

The study is conducted very well and data is analyzed and presented clearly. But, the major concern is difficulty in reading the manuscript due to poor English proficiency.  Highly recommend the manuscript be edited extensively by an English professional for further consideration.

  1. We agree with the reviewer about the poor English proficiency. The article has been edited by ISMETT language service, as reported in the acknowledgement: “Acknowledgments: The authors thank ISMETT’s Language Services Department for his language editing of the manuscript.”

Reviewer 2 Report

In this article, Monica and colleagues examine IgA, IgG, and T-cell responses in solid organ transplant recipients (SOTRs) following the second and third dose of either Pfizer-BioNTech or Moderna COVID-19 vaccine. Importantly, the authors show that the third dose of the vaccine induced improved antibody responses against SARS-CoV-2. Moreover, they demonstrate that SOTRs who contract a SARS-CoV-2 infection following the third dose of vaccine failed to elicit an anti-spike-S1 IgA response both prior to and preceding SARS-CoV-2 infection. This result suggests that anti-SARS-CoV IgA responses might be a valuable indicator of vaccine efficacy, which could apply to the general population and to high-risk groups such solid organ transplant recipients. In general, the article is well-written, and the data presented are clear.

Specific comments

1.     In table 2 it appears that low T-cell responses correlate with severe/fatal SARS-CoV-2 infections, in addition to low IgG levels. However, did low IgA levels also correlate with severe disease within the SARS-CoV-2 infected population? Could the authors expand on this observation and discuss these potential correlations.  

2.     A limitation of the study is the heterogeneity of the vaccines analyzed, the heterogeneity of the immunosuppressive treatments, and the small number of patients examined. However, the data supporting their primary claims that the third dose increases IgG and T-cell responses, and that low IgA levels are associated with contracting SARS-CoV-2 infection appear valid despite these limitations.

Author Response

In this article, Monica and colleagues examine IgA, IgG, and T-cell responses in solid organ transplant recipients (SOTRs) following the second and third dose of either Pfizer-BioNTech or Moderna COVID-19 vaccine. Importantly, the authors show that the third dose of the vaccine induced improved antibody responses against SARS-CoV-2. Moreover, they demonstrate that SOTRs who contract a SARS-CoV-2 infection following the third dose of vaccine failed to elicit an anti-spike-S1 IgA response both prior to and preceding SARS-CoV-2 infection. This result suggests that anti-SARS-CoV IgA responses might be a valuable indicator of vaccine efficacy, which could apply to the general population and to high-risk groups such solid organ transplant recipients. In general, the article is well-written, and the data presented are clear.

We thanks very much the reviewer for the positive comments.

I wish to report that the manuscript is by Miele et al. (Miele is the surname)

Specific comments

  1. In table 2 it appears that low T-cell responses correlate with severe/fatal SARS-CoV-2 infections, in addition to low IgG levels. However, did low IgA levels also correlate with severe disease within the SARS-CoV-2 infected population? Could the authors expand on this observation and discuss these potential correlations.  
  2. We thanks the reviewer for the useful suggestion. We added the column of IgA on table 2, and we discuss the correlation between IgA and disease severity in the results (paragraph 3.5) as follow: “Concerning spike-specific IgA, we reported a positive serology in only 2 non-hospitalized recovered patients, which also presented positive IgG and T cell responses” (written in red in the text)

  1. A limitation of the study is the heterogeneity of the vaccines analyzed, the heterogeneity of the immunosuppressive treatments, and the small number of patients examined. However, the data supporting their primary claims that the third dose increases IgG and T-cell responses, and that low IgA levels are associated with contracting SARS-CoV-2 infection appear valid despite these limitations.

  1. We agree with the reviewer about the limitation of the study that we, however, cited in the discussion (line 342-344)

Round 2

Reviewer 1 Report

Initial comments and concerns are sufficiently addressed.